# Pharmacogenomic Profile of Amazonian Amerindians

**DOI:** 10.3390/jpm12060952

**Published:** 2022-06-10

**Authors:** Juliana Carla Gomes Rodrigues, Marianne Rodrigues Fernandes, André Maurício Ribeiro-dos-Santos, Gilderlanio Santana de Araújo, Sandro José de Souza, João Farias Guerreiro, Ândrea Ribeiro-dos-Santos, Paulo Pimentel de Assumpção, Ney Pereira Carneiro dos Santos, Sidney Santos

**Affiliations:** 1Núcleo de Pesquisa em Oncologia, Universidade Federal do Pará, Belem 66073-000, Brazil; fernandesmr@yahoo.com.br (M.R.F.); akelyufpa@gmail.com (Â.R.-d.-S.); assumpcaopp@gmail.com (P.P.d.A.); npcsantos@yahoo.com.br (N.P.C.d.S.); sidneysantosufpa@gmail.com (S.S.); 2Laboratório de Genética Humana e Médica, Universidade Federal do Pará, Belem 66075-110, Brazil; andremrsantos@gmail.com (A.M.R.-d.-S.); gilderlanio@gmail.com (G.S.d.A.); joao.guerreiro53@gmail.com (J.F.G.); 3BioME, Universidade Federal do Rio Grande do Norte, Natal 59078-400, Brazil; sandro@neuro.ufr.br

**Keywords:** Native American, pharmacogenetics, exome sequencing, pharmacogenes, precision medicine

## Abstract

Given the role of pharmacogenomics in the large variability observed in drug efficacy/safety, an assessment about the pharmacogenomic profile of patients prior to drug prescription or dose adjustment is paramount to improve adherence to treatment and prevent adverse drug reaction events. A population commonly underrepresented in pharmacogenomic studies is the Native American populations, which have a unique genetic profile due to a long process of geographic isolation and other genetic and evolutionary processes. Here, we describe the pharmacogenetic variability of Native American populations regarding 160 pharmacogenes involved in absorption, distribution, metabolism, and excretion processes and biological pathways of different therapies. Data were obtained through complete exome sequencing of individuals from 12 different Amerindian groups of the Brazilian Amazon. The study reports a total of 3311 variants; of this, 167 are exclusive to Amerindian populations, and 1183 are located in coding regions. Among these new variants, we found non-synonymous coding variants in the *DPYD* and the *IFNL4* genes and variants with high allelic frequencies in intronic regions of the *MTHFR, TYMS, GSTT1*, and *CYP2D6* genes. Additionally, 332 variants with either high or moderate (disruptive or non-disruptive impact in protein effectiveness, respectively) significance were found with a minimum of 1% frequency in the Amazonian Amerindian population. The data reported here serve as scientific basis for future design of specific treatment protocols for Amazonian Amerindian populations as well as for populations admixed with them, such as the Northern Brazilian population.

## 1. Introduction

Adverse drug reaction (ADR) may be defined as “any response to a drug which is noxious and unintended, and which occurs at doses normally used in man for prophylaxis, diagnosis, or therapy of disease, or for the modification of physiological function” [1].

ADRs are important cause of patient morbidity and mortality, as they are responsible for increased health costs and lead to lack of patient adherence to medical treatment worldwide. It is estimated they may occur in over 30–60% of drug prescriptions [2].

There are several factors that trigger the appearance of ADRs in different types of treatment schemes, including patient age, disease status, physiological conditions, drug dosage, and drug–drug interaction [3]. Additionally, interindividual genetic variability has also being demonstrated to play an essential role to this condition [4,5].

Pharmacogenomics—the study of genetic variants that may influence drug efficacy, response, and/or toxicity—accounts for around 80% of the variability in drug pharmacokinetics and pharmacodynamics and has been associated with over 60% of ADRs events nowadays [2]. Given this close relationship, an assessment about the pharmacogenomic profile of patients prior to drug prescription or dose adjustment is a pivotal strategy.

The goal of pharmacogenomics is the establishment of precision medicine, increasing drug efficacy and safety while minimizing events of resistance, toxicity, and adverse effects associated to drugs [6]. To achieve this purpose, it is necessary to obtain accurate information about actionable pharmacogenetic variants. The interethnic variability is an important issue to be considered in pharmacogenomic studies since there is an enormous variability in the distribution and frequency of pharmacogenetics variants among different populations, with most of them being rare or specific to a given biogeographical population [3,7,8].

Most of the information acquired in the pharmacogenomic field, especially that employing large-scale sequencing data and specific pharmacogenomic protocols, has been centered on more homogeneous populations, specifically those of European origins [9,10], which cannot be fully applicable to other ethnic groups

Genetically distinct populations need specific algorithms for drug dosing. This is especially relevant when we consider Native American populations, which have a unique genetic profile due to a long process of geographic isolation and other genetic and evolutionary processes, such as genetic drift, founding effect, natural selection, and inbreeding [11].

These populations are extremely underrepresented in genetic research. In parallel, some studies have already demonstrated that Amerindian groups have distinct genetic profiles concerning pharmacogenomic biomarkers widely used in clinical practice [12,13], which further reinforces the need for their inclusion in pharmacogenomic studies.

Therefore, the purpose of this study is to describe the pharmacogenetic variability of Native American populations regarding 160 pharmacogenes involved in absorption, distribution, metabolism, and excretion (ADME) processes and biological pathways of different therapies based on data obtained through complete exome sequencing of 64 individuals from different Amerindian groups of the Brazilian Amazon.

## 2. Materials and Methods

### 2.1. Study Population

The study population is composed of 64 Amerindians. The Amerindians represent 12 different Amazonian ethnic groups that were grouped together as the Native American (NAM) group. Ancestry analysis was performed as described by Ramos et al. 2016 [14], using 61 autosomal ancestry informative markers (AIMs). The individual proportions of European, African, and Native Americans’ genetic ancestries in the NAM group were 0.022, 0.014, and 0.964, respectively. Details such as name, location, and number of individuals in each ethnic group are described in Appendix A. The present study was approved by the National Committee for Ethics in Research (CONEP), identified by Nos. 1062/2006 and 123/98, and the Research Ethics Committee of the UFPA Tropical Medicine Center, under CAAE number 20654313.6.0000.5172. All participants signed a free-informed consent as well as the tribe leaders when necessary. We compared our results with those of populations from other continents obtained from the Exome Aggregation Consortium (ExAC) (available at https://gnomad.broadinstitute.org/, accessed on 6 March 2022). This database is composed of a set spanning 60,706 individuals, including 5203 of African (AFR), 5789 of Latin (AMR), 4327 of East Asian (EAS), 33,370 of European (EUR), and 8256 of South Asian (SAS) descent.

### 2.2. Exome Library Preparation

The exome library preparation of the initial 58 individuals of the NAM group was performed as previously described in [15]. Afterward, six additional individuals were added in this group set, and the exome library preparation is described in [16].

### 2.3. Read Calling and Processing

The quality of the FASTQ reads was analyzed (FastQC v.0.11—http://www.bioinformatics.babraham.ac.uk/projects/fastqc/), and the samples were filtered to eliminate low-quality readings (fastx_tools v.0.13—http://hannonlab.cshl.edu/fastx_toolkit/). The average quality score of reads above 20× and above 10× of coverage were 77% and 89%, respectively. The sequences were mapped and aligned with the reference genome (GRCh38) using the BWA v.0.7 tool (http://bio-bwa.sourceforge.net/). The percentage of reads aligned on the human genome was 85%. Following this alignment with the reference genome, the file was indexed and sorted (SAMtools v.1.2—http://sourceforge.net/projects/samtools/). Subsequently, the alignment was processed for duplicate PCR removal (Picard Tools v.1.129—http://broadinstitute.github.io/picard/), mapping quality recalibration, and local realignment (GATK v.3.2—https://www.broadinstitute.org/gatk/). Of the total number of pairs sequenced, 15% were PCR duplicates and therefore were removed. The results were processed to determine the variants from the reference genome (GATK v.3.2).

### 2.4. Selection of PGx Biomarkers

From the generated exome data, 160 pharmacogenes associated with the ADME process or the pharmacodynamic changes of different drug classes were selected for analysis (Appendix A). The genes were chosen because they are classified as very important pharmacogenes (VIPs) genes by the PharmGKB database [17] and/or because they are described with clinical significance by the Clinical Pharmacogenetics Implementation Consortium (CPIC) guidelines [18]. Additionally, some of these markers are already recommended by drug regulatory agencies as predictors of toxicity and response to therapeutic regimens routinely used in clinical practice.

### 2.5. Bioinformatic and Statistical Analyses

The analysis of the variant annotations was run in the ViVa^®^ (Viewer of Variants) software developed by the Federal University of Rio Grande do Norte (UFRN) bioinformatics team. ViVa^®^ performs genomic annotation for each variant, including chromosomal location and position, reference and variant allele(s), and quality control analysis of nucleotide identification according to Phred score (Q score) (ILLUMINA, 2011) and sample coverage. According to the different National Center for Biotechnology Information (NCBI) databases (https://www.ncbi.nlm.nih.gov/), gene name, RefSeq sequence, gene region, gene number, exon, variant type, amino acid change caused in the protein, and codon change in the DNA sequence were also obtained.

Additionally, annotations also include variant impact according to SnpEFF (http://snpeff.sourceforge.net/), amino acid change caused in the protein, codon change in DNA sequence, polymorphism identification number (rs) according to dbSNP (https://www.ncbi.nlm.nih.gov/snp/), allele frequencies found in ExAC populations, clinical significance according to ClinVar (https://www.ncbi.nlm.nih.gov/clinvar/), reference number cataloged in the COSMIC database (https://cancer.sanger.ac.uk/cosmic), and pathogenicity prediction according to 10 in silico tools (SIFT, PolyPhen, MetalLR, LRT, MAssessor, MTaster, FATHMM, PROVEAN, MetalSVM, and MetaLR).

The allele frequencies of the NAM populations were obtained directly by gene counting and compared with the other study populations (AFR, EUR, AMR, EAS, and SAS). The difference in frequencies between the populations were analyzed by Fisher’s exact test, and results were considered significant when Bonferroni-adjusted *p*-value ≤ 0.00015. The principal component analysis (PCA) was performed using the FactoMineR, version 2.4 package [19]. For all analyses, R package v. 4.1.3 [20] and Python 3.10.4 [21] were used.

## 3. Results

Of the total of 160 genes analyzed, our investigation found a total of 3311 variants in the study subjects. Of this total, 167 are exclusive to Amerindian populations. The general characteristics of the variants are described in Table 1.

The great majority are single-nucleotide variants (SNV). There are six gene regions where the variants were located; most of them were in intronic regions (1567), followed by exonic regions, specifically in the coding sequence DNA (CDS) (1183). Regarding the in silico functional impact prediction, most of the variants received modifier status, which is defined as “generally non-coding variants or variants that affect non-coding genes, where predictions are difficult or there is no evidence of impact” [22].

Among the exclusive variants found in the Amerindian group, we can highlight 21 that presented a population frequency above 0.01 (Table 2). Of these, two are missense-like variants, which have a moderate impact: the variant at position 19:39248563 (MAF = 0.0161) found in the IFNL4 gene and the variant at position 1:97098596 (MAF = 0.0313) in the DPYD gene. In addition, we could also find seven variants with high allelic frequencies (MAF > 0.5) in the Amerindian populations: two in the GSTT1 gene, three in the CYP2D6 gene, one in the MTHFR gene, and one in the TYMS gene. All seven have a modifying impact and are present in intronic regions or 5’UTR regions, which are known to regulate gene expression.

Due to the large number of variants found, we narrowed down variants based in three main criteria: (i) a minimum of 10 reads of coverage; (ii) those which show a high (a disruptive impact in the protein, probably causing protein truncation, loss of function, or triggering nonsense mediated decay) or moderate (a non-disruptive variant that might change protein effectiveness) impact; and (iii) variants that show allelic frequency >1% in at least of two other continental populations. Therefore, in the posterior analyses, the results are based on 332 variants (Appendix A).

Plotting a principal component analysis based on the allelic frequencies of the 332 variants found in each of the six populations (AFR, AMR, EAS, EUR, NAM, and SAS) could explain nearly 66% of the total variation found in our group of variants (Figure 1).

The NAM population formed a cluster isolated from all other study populations, which allows us to corroborate the differentiated genetic profile of this group concerning the variants analyzed in this study. Furthermore, we can observe that the population with the most similar genetic profile of the NAM population is the AMR population, composed of Latin American population groups, followed by the EAS population, formed by population groups from East Asia.

Additionally, the 332 pharmacogenomic variants were evaluated in pairwise comparisons between the NAM population and each of the Exome Aggregation Consortium (ExAC) population, individually, using Fisher’s exact test (Appendix A). The largest number of variants with significant differences was found in the comparison between NAM and AFR (144 variants in total), followed by the comparison NAM × EUR (133). The smallest number of variants with significant differences was observed between the comparison of NAM and AMR (75). The number of variants with significant differences between NAM and Asian populations was basically the same (114 compared with EAS and 115 with SAS). These data agree with that shown in Figure 1. Table 3 shows the 42 pharmacogenetic variants that showed significant differences between NAM and all EXAC populations.

Of these, ten variants are classified with a high impact: rs881711 of the *CBR3* gene; rs200088269rs35616319, rs753045406, and rs757139064 of the *HLA-DRB1* gene; rs199556640 and rs9282026 of the *HLA-DQA1* gene; rs74597329 and rs11322783 of the *IFNL4* gene; rs3745274 of the *CYP2B6* gene; and finally, rs149012039 of the *CYP2D6* gene. Additionally, most of variants present higher allele frequencies in the group of Amazonian Amerindians compared to the other world populations except for nineteen variants that present the opposite profile (rs881711, rs1050147, rs130066, rs149012039, rs11322783, rs2032582, rs396991, rs3745274, rs4963, rs707952, rs2308466, rs2523600, rs1042713, rs713031, rs697742, rs1143146, rs1131170, rs2228001, and rs2228570).

## 4. Discussion

The incorporation of pharmacogenomic assessment prior to treatment would be of great benefit in terms of costs, quality of life, and optimization of therapeutic resources. Given this, one of the main goals in the future of pharmacogenomic research is characterization of gene polymorphisms to elucidate the genetic background underlying differences in drug responses. However, some population groups remain underrepresented in these types of investigations, such as Native American populations. Recently, some efforts have been made to include genetic information about allelic frequencies of Native American populations and their influence on susceptibility to diseases, such as severe forms of COVID-19 [23], psychological disorders [24], oncological diseases [25,26], and also on different types of pharmacological treatments [12,13,27,28]. Most of these studies are focused on one or a few group of genes. Hence, there is still a paucity of large-genome-scale studies on Native American populations, particularly Amazonian Amerindian groups, regarding the response pattern and toxicity of drugs that are known to have pharmacogenetic influences. The focus of this work was to evaluate genes of pharmacogenomic importance as recommended by global drug regulatory institutions, such as the Food and Drug Administration from United States, the European Medicine Agency from Europe, and the Clinical Pharmacogenetics Implementation Consortium (CPIC), in a group of 64 Amerindians from the Amazon region of Brazil. This is the first study to analyze a large set of guided-treatment pharmacogenes by whole-exome sequencing in Amazonian Amerindian populations.

### 4.1. General Data Found in the Amazonian Amerindian Population

The exome sequencing of the 160 pharmacogenes analyzed revealed more than 3000 variants. Interestingly, most variants that have already been defined as a biomarker-definers of drug indication, such as *DPYD*2A, DPYD *13,* rs67376798, or rs75017182 of the *DPYD* [29] and the rs9923231 of the *VKORC1* genes [30], were not found in the set of Amerindians of this study. An exception is the rs1272632214, which was previously reported [16] with a high frequency in the Amerindians of this study and in complete linkage disequilibrium with the rs116855232 of the *NUDT15* gene, which is a variant linked to thiopurine toxic, included in the recommendations of CPIC guidelines for adjusting starting doses of azathioprine, mercaptopurine, and thioguanine [31].

An important finding of this is study is the large number of novel variants firstly reported in the Amazon Amerindians groups (here, they will be referred by their chromosomal location). Most variants were SNVs, and one-third of them were located in gene coding sequences. Of the 167 novel variants reported, nine must be highlighted: (1) the 19:39248563 of the *IFNL4* gene; (2) the 1:97098596 in the *DPYD* gene; (3) the 22_KI270879v1_alt:278462 and 22_KI270879v1_alt:278129 of the *GSTT1*; (4) the 1:11803345 in *MTHFR*; (5) the 18:657685 in *TYMS*; and (6) the 22_KI270928v1_alt:52910, 22_KI270928v1_alt:52912, and 22_KI270928v1_alt:52919 of the *CYP2D6* gene. To validate the results reported here, these variants must be confirmed in studies with a larger number of individuals and analyzed regarding their functional effect. Here, we will discuss the role of these genes to the pharmacogenomic field.

#### 4.1.1. *IFNL4*

The *IFNL4*, along with three neighboring genes of the interferon lamba family (*IFNL1*, *IFNL2,* and *IFNL3)* leads to activation of the JAK-STAT signaling pathway and upregulation of numerous interferon-stimulated genes [32]. This gene has been included in the CPIC guidelines in association with peginterferon alfa-2a and peginterferon alfa-2b, two regimen treatments with direct-acting antivirals approved for hepatitis C virus (HCV) genotype 1 infection [33]. Variants in the *IFNL4* have shown to be a better predictor of response to PegIFN in patients with either African or European ancestry [34]. Because there is a strong relationship with sustained virologic response to HCV treatments, a polymorphism (rs12979860) of the *IFNL4* gene is included as an actionable variant in drug labels for sofosbuvir, ombitasvir, paritaprevir, and ritonavir by the Swissmedic agency (www.swissmedic.ch, accessed on 3 May 2022).

#### 4.1.2. *DPYD*

The *DPYD* gene encodes the dihydropyrimidine dehydrogenase (DPD), a critical enzyme responsible for catabolism of approximately 85% of the 5-fluorouracil (5-FU) to its inactive metabolic form [35]. The 5-flurouracil is the active agent of fluoropyrimidine chemotherapy, frequently prescribed for treatment of a variety of cancers, including colorectal, upper gastrointestinal, breast, and head and neck cancers. Despite their large use in clinical practice, the fluoropyrimidine treatment has a narrow therapeutic index; even at standard dose, a substantial proportion of patients may develop grade 3 or higher toxicities [36]. The guidelines of the Clinical Pharmacogenetics Implementation Consortium (CPIC) describe four decreased-function *DPYD* variants, namely rs3918290 (also known as *DPYD***2A*) rs55886062 (*DPYD* **13*), rs67376798 (p.D949V), and rs75017182 (HapB3), and strongly recommend the use of alternative drugs or the reduction (25% to 50%) of the standard dose of fluoropyrimidines for patients who are classified as *DPYD* intermediate or poor metabolizer, respectively [29]. A study evaluating Amerindian populations from the Amazon reported a total of nine polymorphic variants of the *DPYD*; two of them are variants responsible for decreasing the DPD enzymatic activity (included in the CPIC guidelines), the rs3918290 and rs55886062, with 1% and 2% of frequency, respectively; and three of them (rs17116806, rs1801159, and rs4970722) had frequencies higher than 40%, contrasting with frequencies much lower found in populations with African, European, or Asian ancestry; these variants show lower frequencies [12].

#### 4.1.3. *GSTT1*

The *GSTT1* belong to the super family of glutathione S-transferases. These enzymes are implicated in the second-phase metabolism of several compounds, such as anticancer and xenobiotic substances [37]. Polymorphisms in the *GSTT1* gene show significant frequency differences among distinct ethnic groups: East Asians present the highest frequencies worldwide for *GSTT1* deletions [38], which could implicate in higher interindividual variability in pharmacotherapy responses and susceptibility to various diseases, especially those of hepatological disorders [39]. The results regarding the role of *GSTT1* conflict depending on the type of treatment evaluated. Regarding breast cancer therapy, the *GSTT1* double-null genotype has been significantly associated with an increased tumor response [40], while for patients with chronic myeloid leukemia, this genotype has been linked to worse treatment outcome to imatinib [41].

#### 4.1.4. *MTHFR*

The *MTHFR* gene encodes the 5,10-methylenetetrahydrofolates enzyme, a substrate for de novo purine synthesis and therefore crucial to produce amino acids. Additionally, this enzyme is involved in the conversion of homocysteine to methionine, which is then converted to the universal methyl donor, S-adenosylmethionine, used for methylation of DNA and proteins [42]. Given the role of *MTHFR* in both DNA synthesis and methylation status, this gene participates of several pathways commonly used as targets for chemotherapeutic antineoplastic and antirheumatic drugs, such as 5-fluorouracil and methotrexate [43,44]. The *MTHFR* gene has two main variants: the rs1801133 and rs1801131. A recent meta-analysis evaluating 34 studies showed that both variants are statistically associated with toxicities during high-dose methotrexate treatment; the rs1801133 was positively associated with increased risk of hepatotoxicity, mucositis, and renal toxicity, while the rs1801131 showed a decreased risk of renal toxicity during the therapy [45]. Concerning fluoropyrimidine-based treatments, results involving these variants are conflicting: while Campbell and colleagues found significant associations of rs1801133 and rs1801131 with a higher frequency of nonhematologic toxicity (nausea/vomiting), a protective effect for neutropenia, and global toxicity [46], Zhong and collaborators found no significant association between both polymorphisms and the clinical response to fluoropyrimidine-based chemotherapy under any of the three genetic models (allele model, dominant model, and recessive model) [47]. Curiously, a study evaluating patients with gastrointestinal or colorectal cancer from the Brazilian Amazon region in which the study population showed means of 31.1% of Amerindian ancestry, reported a significant association with the rs1801133 in the *MTHFR* and severe toxicity during fluoropyridimine-based treatment [48].

#### 4.1.5. *TYMS*

The *TYMS* gene encodes the thymidylate synthase, the enzyme responsible for catalyzing the methylation of deoxyuridylate (dUMP) to deoxythymidylate (dTMP). Since the dTMP is the only source of intracellular thymidine, the TS is a key enzyme for the production of DNA and the principal target of 5-fluorouracil [49]. The expression levels of TS are crucial to determine the effectiveness of 5-FU because the inhibition of TS can disrupt normal DNA synthesis, leading tumor cells to apoptosis. Studies have shown the downregulation of *TYMS* contributes to 5-FU sensitivity [50], while the overexpression of TS represents a pathway of tumor cells resistance to 5-FU [51]. One of the main reasons for increased TS levels in cancer cells is polymorphisms in the *TYMS* gene, particularly a triple tandem repeat (*TSER *3*) detected in the 5′-UTR of the gene; also, TS copy number and different genomic instability statuses, such as chromosome and microsatellite instability, may influence 5-FU responsiveness [52]. The FDA-approved starting dosage for capecitabine monotherapy (a prodrug of 5-FU) is 1250 mg/m^2^ b.i.d on a 2 weeks on/1 week off regimen; however, a clinical trial conducted by Soo and colleagues suggested that patients with the *TSER 3R/3R* genotype exhibit enhanced tolerance to capecitabine and may potentially benefit from a higher dose (1500 mg/m^2^) compared to the standard protocol of 1250 mg/m^2^ [53]. Another study showed that the overall survival of patients with high TS levels observed in primary tumors with metastasis and in those with lymph node metastasis is shorter than in patients with low TS levels [52]. Together, these findings suggest that the *TYMS* genotyping assessment may predict efficacy and tolerance of 5-FU-based treatment.

#### 4.1.6. *CYP2D6*

The cytochrome P450 2D6 (*CYP2D6*) is responsible for metabolizing approximately 25% of all drugs in the human liver [54,55]. Typical CYP2D6 substrates are lipophilic bases, including some anticancer drugs, antidepressants, antipsychotics, antiarrhythmics, antiemetics, beta-adrenergic receptor antagonists (beta-blockers), and opioids. At least 160 therapeutic targets are known to be metabolized by the CYP2D6 enzyme. Of all the pharmacogenetic biomarkers categorized by the FDA, there are 69 drugs from different pharmacological classes that have relevant pharmacogenomic information regarding the *CYP2D6* gene. The information ranges from simple descriptions of clinical pharmacology and drug interactions to even possible dose adjustments depending on the *CYP2D6* genotype. The CPIC also formulated six specific guidelines for genetic testing regarding possible dose adjustments based on the metabolism profile of *CYP2D6*, which include: (i) anesthetics [56], specifically codeine, which is bioactivated in morphine by *CYP2D6*; (ii) selective serotonin reuptake inhibitor drugs [57], which are the main treatment options for depressive and anxiety disorders; (iii) tricyclic antidepressants [58], for example, nortriptyline, desipramine, etc.; (iv) 5-hydroxytryptamine type 3 (5-HT3) receptor antagonists [59] used for the prevention of nausea and vomiting induced by chemotherapy, radiation, and postoperative care; (v) tamoxifen, a selective estrogen receptor modulator [60] used against breast cancer; and (vi) atomoxetine [61], a non-stimulant medication used to treat attention-deficit/hyperactivity.

### 4.2. Comparative Analyses between Variants Found in the Amazon Amerindians and the Five Continental Populations from ExAC Database

The origin and expansion of Amerindian populations in the Amazon have been the subject of numerous academic discussions and still remain challenging. Historical evidence indicates that the primitive populations that arrived in the Amazon derive from an ancestral population that originally left Asia by way of the Bering Strait more than 15,000 years ago [62,63]. Other analyses have also shown that some groups in South America, particularly in Brazil, share alleles with indigenous New Guineans, Australians, and Andaman Islanders, which together form a group with a distinct genomic profile known as Australasians [64,65,66].

Recently, Ribeiro-dos Santos et al., 2020, contributed to the reconstruction of the genetic history of Native Brazilian populations, performing whole-exome sequencing of the 58 Amazonian Native Americans included in this study and confirming an occupation model with separate migration waves in the region [15]. These data confirm the unique Amazonian genetic signature in the Amerindian populations, which may also be observed in the PCA generated in this study through the complete segregation of the Amazonian Amerindians from the other populations.

The most closely plotted groups of Amazonian Amerindians were individuals from the AMR group that share origin branches with Native Americans and individuals from East Asia, the ancestral group that engendered the America continent settlement. Still, the PCA also shows an isolated cluster formed by the African population. These findings agree with the history of human evolution: Amerindians are more genetically similar to American and East Asian populations and more genetically distant from African populations [67,68]. This pattern of genetic profile between Amazonian Amerindians and other global population corroborates with other genetic association studies [12,24,25].

## 5. Conclusions

Our study analyzed genes of great importance in the pharmacogenomics of different types of therapies, based in data generated from whole-exome sequencing of a group of Amazonian Amerindians from northern Brazil. Our results identified many novel variants located in essential genes to cancer (*DPYD, TYMS, MTHFT*, and *GSTT1*), infectious diseases (*IFNL4*), and psychiatric treatments (*CYP2D6*). This study also demonstrates that the Amazonian Amerindians show very distinct allelic frequencies of pharmacogenomic variants when compared to populations of African, European, Asians, and even Latin origins, which could be observed by statistical and clustering comparative analysis.

These data reinforce the distinct genetic profile concerning biomarkers widely used in clinical practice in Amazonian Amerindians, which further reinforces the need for their inclusion in pharmacogenomic research. Studies with a larger number of individuals and assessment of functional impact of the variants unraveled here are needed to confirm our results. Together, these findings may serve as the scientific basis for the future design of specific treatment protocols for Amazonian Amerindian populations as well as for populations admixed with them, such as the northern Brazilian population.

## Figures and Tables

**Figure 1 jpm-12-00952-f001:**
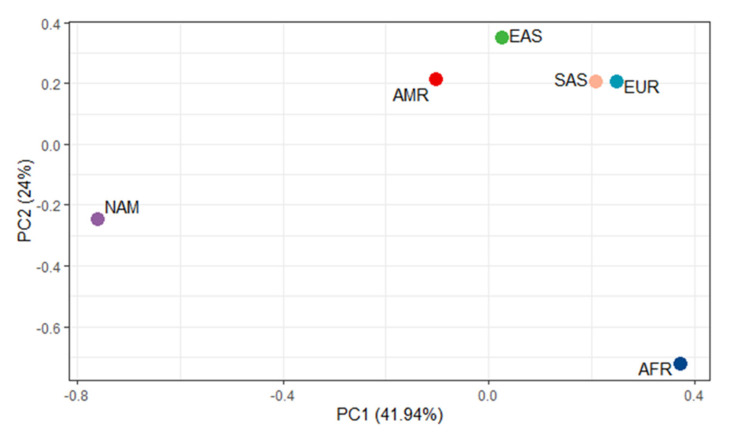
Principal component analysis of the 332 pharmacogenetic variants analyzed in the six populations of the present study. AFR, African; AMR, American; EAS, East Asian; EUR, European; NAM, Amazonian Amerindian; SAS, South Asian.

**Table 1 jpm-12-00952-t001:** Description of all variants found in the exome of 64 Amazonian Amerindians of the study.

Characteristics	Number
All variants	3.311
New variants	167
Type	
Single-nucleotide variant (SNV)	2.934
Insertion or deletion (INDEL)	377
Gene region	
3′UTR	188
5′UTR	90
Exon (CDS)	1183
Intragenic	33
Intron	1567
Others	250
Consequence	
Codon change and codon deletion	10
Codon change and codon deletion + splice site region	1
Codon change and codon insertion	2
Codon deletion	5
Codon deletion + splice site region	1
Codon Insertion	2
Frameshift	53
Frameshift + splice site region + splice site donor	1
Intragenic	33
Intronic	1567
Protein functional site	82
Nonsynonymous	641
Protein structural interaction locus	30
Splice site acceptor	3
Splice site donor	8
Splice region	102
Start codon	1
Stop codon	5
Synonymous	486
3′UTR	188
5′UTR	90
SNPeff Impact	
High	101
Moderate	672
Modifier	1878
Low	660

**Table 2 jpm-12-00952-t002:** Novel variants reported in the Amazonian Amerindians with minor allele frequency greater than 1%.

Chromosomal Position	Reference	Variant	Gene	Region Detailed	Impact	Minor Allele Frequency
19:39248563	C	G	*IFNL4*	NON_SYNONYMOUS_CODING	MODERATE	0.0161
X:134500131	G	T	*HPRT1*	UTR_3_PRIME	MODIFIER	0.0161
19:38442623	G	T	*RYR1*	INTRON	MODIFIER	0.0161
X:38352846	G	C	*OTC*	INTRON	MODIFIER	0.0161
1:186679301	G	A	*PTGS2*	INTRON	MODIFIER	0.0161
7:99778057	T	C	*CYP3A4*	SYNONYMOUS_CODING	LOW	0.0172
4:20850609	C	T	*KCNIP4*	SYNONYMOUS_CODING	LOW	0.0172
15:74754842	C	A	*CYP1A2*	SYNONYMOUS_CODING	LOW	0.0172
X:154534297	AC	A	*G6PD*	INTRON	MODIFIER	0.0208
10:112951639	TGCCC	T	*TCF7L2*	INTRON	MODIFIER	0.0250
9:130855098	TAGGGG	T	*ABL1*	SPLICE_SITE_REGION + INTRON	LOW	0.0250
1:97098596	T	C	*DPYD*	NON_SYNONYMOUS_CODING	MODERATE	0.0313
2:233760783	C	T	*UGT1A4*	INTRON	MODIFIER	0.0357
2:233760783	C	T	*UGT1A1*	SYNONYMOUS_CODING	LOW	0.0357
22_KI270879v1_alt:278462	C	G	*GSTT1*	UTR_5_PRIME	MODIFIER	0.0500
1:11803345	C	A	*MTHFR*	INTRON	MODIFIER	0.0556
18:657685	G	GGCCTGCCTCCGTCCCGCCGCGCCACTTC	*TYMS*	UTR_5_PRIME	MODIFIER	0.1154
22_KI270928v1_alt:52910	C	G	*CYP2D6*	INTRON	MODIFIER	0.5400
22_KI270879v1_alt:278129	C	T	*GSTT1*	INTRON	MODIFIER	0.5417
22_KI270928v1_alt:52912	T	G	*CYP2D6*	INTRON	MODIFIER	0.6230
22_KI270928v1_alt:52919	G	C	*CYP2D6*	INTRON	MODIFIER	0.6563

**Table 3 jpm-12-00952-t003:** Pairwise comparison of variants with a statistically significant difference of allele frequency between Amazonian Amerindians and all continental populations from the ExAC database.

			Pairwise Comparation (*p*-Value)				
Chromosomal Location	Gene	dbSNP	NAM × AFR	NAM × AMR	NAM × EAS	NAM × EUR	NAM × SAS
chr7:87531302	*ABCB1*	rs2032582	2.18306 × 10^−56^	6.53986 × 10^−16^	2.51344 × 10^−12^	1.02973 × 10^−16^	1.00647 × 10^−7^
chr4:2915035	*ADD1*	rs4963	9.45593 × 10^−5^	2.69477 × 10^−6^	4.94438 × 10^−17^	3.89984 × 10^−5^	3.67996 × 10^−5^
10:114045297	*ADRB1*	rs1801253	3.97318 × 10^−15^	0.000145471	2.41152 × 10^−8^	1.2311 × 10^−9^	4.82047 × 10^−8^
chr5:148826877	*ADRB2*	rs1042713	1.19482 × 10^−17^	6.47618 × 10^−14^	1.1541 × 10^−20^	3.6564 × 10^−12^	8.98985 × 10^−16^
chr11:113400106	*ANKK1*	rs1800497	8.91338 × 10^−13^	1.89293 × 10^−7^	7.07468 × 10^−10^	1.75701 × 10^−25^	8.89022 × 10^−17^
chr11:113396099	*ANKK1*	rs7118900	1.57472 × 10^−13^	4.5083 × 10^−7^	1.35542 × 10^−9^	2.61132 × 10^−26^	2.51963 × 10^−16^
chr22:23285051	*BCR*	rs12484731	2.18658 × 10^−26^	1.16413 × 10^−6^	9.95701 × 10^−26^	5.38079 × 10^−46^	2.00229 × 10^−35^
chr22:23290360	*BCR*	rs35537221	1.8611 × 10^−28^	1.05573 × 10^−6^	4.00474 × 10^−21^	2.14428 × 10^−45^	1.21405 × 10^−35^
chr21:36135447	*CBR3*	rs881711	5.8767 × 10^−19^	2.17402 × 10^−19^	4.67622 × 10^−28^	2.69704 × 10^−29^	1.15895 × 10^−27^
chr6:31154538	*CCHCR1*	rs130066	6.77941 × 10^−5^	1.33884 × 10^−6^	0.000115616	1.58769 × 10^−6^	4.19502 × 10^−5^
chr19:41006936	*CYP2B6*	rs3745274	1.64912 × 10^−11^	8.995 × 10^−10^	5.73451 × 10^−5^	9.859 × 10^−7^	1.34727 × 10^−12^
chr22:42142513	*CYP2D6*	rs149012039	1.14215 × 10^−11^	1.59914 × 10^−6^	6.78214 × 10^−20^	3.22875 × 10^−9^	7.70391 × 10^−5^
chr1:161544752	*FCGR3A*	rs396991	7.44824 × 10^−9^	1.65936 × 10^−5^	7.84237 × 10^−9^	1.16585 × 10^−9^	2.03133 × 10^−9^
chr6:29943337	*HLA-A*	rs3173420	4.31224 × 10^−9^	6.87922 × 10^−8^	4.56562 × 10^−7^	8.57084 × 10^−12^	8.71629 × 10^−12^
chr6:29942965	*HLA-A*	rs281864739rs78306866	8.46059 × 10^−12^	4.05707 × 10^−17^	8.97765 × 10^−12^	3.63056 × 10^−14^	4.43796 × 10^−13^
chr6:29942953	*HLA-A*	rs199474436	5.79981 × 10^−6^	6.93234 × 10^−13^	1.41018 × 10^−11^	6.78811 × 10^−9^	6.08387 × 10^−6^
chr6:29942581	*HLA-A*	rs1143146	2.45192 × 10^−16^	3.07371 × 10^−24^	1.35824 × 10^−18^	1.34049 × 10^−18^	9.97823 × 10^−14^
chr6:31356226	*HLA-B*	rs2308466	3.65704 × 10^−9^	3.66621 × 10^−9^	5.81378 × 10^−5^	2.5862 × 10^−10^	7.79316 × 10^−12^
chr6:31356227	*HLA-B*	rs2523600	1.9236 × 10^−13^	2.13422 × 10^−14^	8.63059 × 10^−15^	4.81953 × 10^−11^	7.03642 × 10^−13^
chr6:31356889	*HLA-B*	rs713031	1.07572 × 10^−17^	1.41509 × 10^−16^	6.48134 × 10^−29^	2.226 × 10^−13^	1.45874 × 10^−21^
chr6:31356247	*HLA-B*	rs697742	1.16978 × 10^−17^	4.25927 × 10^−20^	3.31496 × 10^−24^	2.21922 × 10^−16^	3.51264 × 10^−21^
chr6:31356928	*HLA-B*	rs1131170	2.28835 × 10^−24^	1.15345 × 10^−18^	2.11729 × 10^−29^	4.93973 × 10^−20^	6.05278 × 10^−27^
chr6:31269347	*HLA-C*	rs1130838	6.45396 × 10^−16^	2.25074 × 10^−12^	1.6135 × 10^−15^	4.15458 × 10^−9^	5.96131 × 10^−12^
chr6:31271999	*HLA-C*	rs2074493	8.10462 × 10^−10^	9.63528 × 10^−11^	1.02002 × 10^−10^	9.09483 × 10^−15^	4.07089 × 10^−20^
chr6:31270025	*HLA-C*	rs1050147	1.18398 × 10^−31^	3.7936 × 10^−27^	7.42085 × 10^−32^	7.33005 × 10^−23^	1.50973 × 10^−26^
chr6:33080851	*HLA-DPB1*	rs1042136	1.6523 × 10^−8^	1.88517 × 10^−7^	2.47857 × 10^−10^	5.23217 × 10^−5^	1.23339 × 10^−6^
chr6:32641487	*HLA-DQA1*	rs1142333	4.373 × 10^−28^	1.9016 × 10^−20^	3.07937 × 10^−28^	1.11648 × 10^−30^	4.88343 × 10^−43^
chr6:32641535	*HLA-DQA1*	rs1129808	7.97358 × 10^−92^	1.19523 × 10^−84^	8.96057 × 10^−84^	8.9906 × 10^−104^	3.2328 × 10^−109^
chr6:32641477	*HLA-DQA1*	rs1142331	1.18138 × 10^−25^	7.36406 × 10^−16^	2.39546 × 10^−25^	2.60828 × 10^−27^	1.4441 × 10^−39^
chr6:32641521	*HLA-DQA1*	rs199556640	9.45834 × 10^−19^	2.68598 × 10^−19^	3.99666 × 10^−22^	6.58968 × 10^−19^	1.29641 × 10^−24^
chr6:32641519	*HLA-DQA1*	rs9282026	6.76015 × 10^−12^	5.23003 × 10^−12^	1.15636 × 10^−14^	8.59051 × 10^−12^	3.46752 × 10^−16^
chr6:32642029	*HLA-DQA1*	rs707952	7.78654 × 10^−6^	4.53726 × 10^−8^	9.34217 × 10^−5^	3.23443 × 10^−7^	2.9079 × 10^−7^
chr6:32581807	*HLA-DRB1*	rs200088269rs35616319	7.84356 × 10^−15^	1.98563 × 10^−8^	7.60999 × 10^−12^	5.08348 × 10^−10^	2.14344 × 10^−10^
chr6:32581802	*HLA-DRB1*	rs753045406	1.2253 × 10^−13^	9.15172 × 10^−8^	4.74343 × 10^−11^	3.01143 × 10^−9^	1.4626 × 10^−9^
chr6:32581621	*HLA-DRB1*	rs757139064	4.04995 × 10^−18^	6.91184 × 10^−18^	2.46801 × 10^−16^	2.08408 × 10^−16^	7.75881 × 10^−20^
chr19:39248515	*IFNL4*	rs74597329	9.56326 × 10^−33^	4.95051 × 10^−6^	1.16957 × 10^−5^	4.53584 × 10^−15^	1.72728 × 10^−6^
chr19:39248513	*IFNL4*	rs11322783	9.56326 × 10^−33^	4.95051 × 10^−6^	1.16957 × 10^−5^	4.53584 × 10^−15^	1.72728 × 10^−6^
chr6:160540105	*LPA*	rs3798220	1.75821 × 10^−48^	7.48752 × 10^−5^	8.65473 × 10^−19^	1.63951 × 10^−43^	6.71261 × 10^−58^
chr12:21178615	*SLCO1B1*	rs4149056	9.76195 × 10^−23^	7.37959 × 10^−10^	2.78132 × 10^−8^	2.34092 × 10^−6^	3.06483 × 10^−17^
chr2:159230143	*TANC1*	rs4664277	1.61934 × 10^−17^	5.62787 × 10^−8^	3.42429 × 10^−10^	1.86855 × 10^−24^	5.11983 × 10^−15^
chr12:47879112	*VDR*	rs2228570	7.39552 × 10^−41^	6.3868 × 10^−19^	1.78553 × 10^−20^	4.04936 × 10^−25^	1.4743 × 10^−38^
chr3:14145949	*XPC*	rs2228001	2.43937 × 10^−34^	1.84077 × 10^−32^	4.51201 × 10^−27^	1.39797 × 10^−23^	1.02515 × 10^−27^

## Data Availability

The data obtained from the public domain are available at gnomAD (broadinstitute.org), and the sequencing data of the Amazonian Amerindian populations are available at the ENA database under the accession number PRJEB35045.

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
