# Peer review of "Pharmacogenomic Profile of Amazonian Amerindians"

_jpm, 2022, doi:10.3390/jpm12060952_

Round 1
Reviewer 1 Report
While this population is understudied, this is a very small sample size for a pharmacogenomics study, and similar studies have been done in this population with similar results. Nevertheless, it is important to add this kind of data to the literature to continue to build an evidence base to help determine allele frequencies in underrepresented populations, and the study design and interpretation overall are sound. Would recommend that the authors add some statements in to the discussion about anything that differentiates their study from others which are similar.
Author Response
Thank you for your consideration about our study, your careful reading, and your suggestion.
We believe that the main difference in our study is the number of genes analyzed, the methodology we used to investigate them and their classification: all of the chosen genes have great importance in pharmacogenomic field.
Therefore, we have modified the discussion according your suggestion. Please see lines 239-247 and lines 251-253.
Reviewer 2 Report
The paper "Pharmacogenomic profile of Amazonian Amerindians" by Gomes Rodrigues and colleages describes the genetic, genomic and pharmacogenomic peculiarities of Native Brazilian Amazon populations, defined by their uniqueness and homogeneity, a result of geographic isolation. The authors adopt an exome-wide sequencing approach and report the variant status, including 167 new Native-specific variants. The paper is truly well written, and shares with thre community a great wealth of genomic information, followed by an intelligent interpretation of the results, which focuses on the the genetic location of the found variants. I think the paper is close to be ready for publication, following a general English language overview (I found a few minor typos and plural/singular issues). Also, a more detailed report on the NGS results should be reported. Sharing their data not only via publication but also via worldwide public databases like GnomAD is a proper strategy as well, for which I applaud the authors. Below, my specific points.
In the abstract, the authors should specify if, whether and how many of the 3,311 variants (including 167 new variants) are located in coding regions (the authors currently list only a few examples on selected genes, such as DPYD and INFNL4).
Abstract and introduction are following a different structure, the earlier focusing on the Amerindian populatio, then specificying the genomic study performed, and its implications for personalized medicine and pharmacogenomics. On the other hand, the introduction focuses on pharmacogenomic aspects, and then talks about the Native American population investigated. I suggest to use the same structure in both sections.
Methods, line 102. The authors describe their NGS pipeline, starting from quality control of reads (they use FastQC, the standard tool), and indicating which tools they used for alignment, et cetera. However, the authors do not report the results of the early quality control. What is the average quality score of the reads? How many PCR duplicates were removed? What's the percentage of reads that aligned on the human genome?
Methods, line 145: PCA is singular, so it "was" performed, not "were".
Methods, line 146: the R software should be cited (there is a recent review on it on the MDPI journal Life), not the IDE Rstudio. Same goes for PyCharm, which is the development environment, but the correct citation and version to report should be that of Python (e.g.https://www.nature.com/articles/s41586-020-2649-2). Reporting the version of the IDE is irrelevant, what makes the pipeline reproducible and the methods section appropriate is the version of R (e.g. 4.1.3) and Python (e.g. 3.3).
Line 382: "remains" should be "remain" as the subject is plural ("The origin and expansion").
Author Response
Thank your for your consideration about our study, your careful reading, and your constructive reviews that greatly helped improve our manuscript. We do intend to share our data via worldwide public databases. Also, we have checked the manuscript and corrected some typos and plural/singular issues throughout the text.
To facilitate, we provided a point-by-point response to your specific points.
In the abstract, the authors should specify if, whether and how many of the 3,311 variants (including 167 new variants) are located in coding regions (the authors currently list only a few examples on selected genes, such as DPYD and INFNL4)
Response: We have modified the abstract accordingly your suggestion. Please see line 26.
Abstract and introduction are following a different structure, the earlier focusing on the Amerindian populatio, then specificying the genomic study performed, and its implications for personalized medicine and pharmacogenomics. On the other hand, the introduction focuses on pharmacogenomic aspects, and then talks about the Native American population investigated. I suggest to use the same structure in both sections.
Response: We have modified the abstract to follow the same structure as our introduction section. Please see lines 16-21.
Methods, line 102. The authors describe their NGS pipeline, starting from quality control of reads (they use FastQC, the standard tool), and indicating which tools they used for alignment, et cetera. However, the authors do not report the results of the early quality control. What is the average quality score of the reads? How many PCR duplicates were removed? What's the percentage of reads that aligned on the human genome?
Response: The average quality score of reads above 20x and above 10x of coverage were 77% and 89%, respectively. The percentage of reads aligned on the human genome was 85%. Of the total number of pairs sequenced, 15% were PCR duplicates, therefore, were removed from posterior analyses. We have added these results of early quality control in the subsetion “2.3. Read calling and processing” in the “Material and Methods” topic. Please see lines 106-118.
Methods, line 145: PCA is singular, so it "was" performed, not "were".
Response: The text has been modified accordingly.
Methods, line 146: the R software should be cited (there is a recent review on it on the MDPI journal Life), not the IDE Rstudio. Same goes for PyCharm, which is the development environment, but the correct citation and version to report should be that of Python (e.g.https://www.nature.com/articles/s41586-020-2649-2). Reporting the version of the IDE is irrelevant, what makes the pipeline reproducible and the methods section appropriate is the version of R (e.g. 4.1.3) and Python (e.g. 3.3).
Response: Thank your for your suggestion. We have added the requested references. Please see references 20 and 21 (lines 519-521).
Line 382: "remains" should be "remain" as the subject is plural ("The origin and expansion").
Response: The text has been modified accordingly.